# Application of Machine Learning in the Control of Metal Melting Production Process

**Nedeljko Dučić** [1],*, **Aleksandar Jovičić** [2], **Srećko Manasijević** [3], **Radomir Radiša** [3], **Žarko Ćojbašić** [4] **and Borislav Savković** [5]

1   Faculty of Technical Sciences Čačak, University of Kragujevac, 32000 Čačak, Serbia
2   Department of Mechanical Engineering, Technical College Čačak, 32000 Čačak, Serbia; aleksandar.jovicic@vstss.com
3   Research and Development Institute Lola Ltd., 11000 Belgrade, Serbia; srecko.manasijevic@li.rs (S.M.); radomir.radisa@li.rs (R.R.)
4   Faculty of Mechanical Engineering, University of Niš, 18000 Niš, Serbia; zcojba@ni.ac.rs
5   Faculty of Technical Sciences, University of Novi Sad, 21000 Novi Sad, Serbia; savkovic@uns.ac.rs
*   Correspondence: nedeljko.ducic@ftn.kg.ac.rs; Tel.: +381-32302733

**Abstract:** This paper presents the application of machine learning in the control of the metal melting process. Metal melting is a dynamic production process characterized by nonlinear relations between process parameters. In this particular case, the subject of research is the production of white cast iron. Two supervised machine learning algorithms have been applied: the neural network and the support vector regression. The goal of their application is the prediction of the amount of alloying additives in order to obtain the desired chemical composition of white cast iron. The neural network model provided better results than the support vector regression model in the training and testing phases, which qualifies it to be used in the control of the white cast iron production.

**Keywords:** metal melting; machine learning; control; smart foundry

## 1. Introduction

The casting process is an old manufacturing technology of which the fundamental principles have not changed to this day. The scientific knowledge about the mentioned production process has been constantly evolving, and as a result, nowadays there are efficient casting processes which make the casting industry diverse and often specialized for a particular type of casting parts. However, a special challenge today is the complete implementation of the idea brought by Industry 4.0 in the foundries. This idea is to build smart foundries, characterized by a centralized system that manages customer orders, supply chains and production activities. The developmental path to the realization of the concept of the smart foundry is long and requires a workforce that understands the importance of the concept, and is ready to monitor the progress in technology. One of the segments of approaching the concept of the smart foundry is the application of artificial intelligence techniques in research, and development of production processes in foundries, as well as their implementation in regular production structures. In the last fifteen years, many researchers have promoted the application of artificial intelligence in the casting process field. Below are provided the main results of the studies in which artificial intelligence techniques have been applied in the following technologies in foundries: the melting metal process, sand casting process, die casting process, continuous casting process and investment casting process. Bouhouche et al. (2004) presented the application of neural networks in the steel industry. Namely, they introduced the control of the melting process by using neural networks, with the goal to optimally control the input variables such as the weights of additives (FeMn,

FeSi, and coke) and heating temperature (T) [1]. Fernandez et al. (2008) developed a neuro-fuzzy model for improving the control through a better prediction of the final temperature and, as a consequence, to reducing the consumption of energy in the electric arc furnace [2]. Anupam et al. (2010) presented the control strategies of an electric arc furnace based on artificial neural networks (ANNs) and an adaptive neuro fuzzy inference system (ANFIS). It involves the prediction of the control action which aids in the reduction of carbon, manganese, and other impurities from the in-process molten steel [3]. Karunakar and Datta (2008) developed an intelligent system based on backpropagation neural networks, with a goal to predict major casting defects such as cracks, misruns, scabs, blowholes and air-locks. Input parameters in the neural network were: green compression strength, green shear strength, permeability, moisture percent, composition of the charge and melting conditions. The research was based on a large set of data, collected in the foundry. The goal of the developed neural network was the preventive action and the prediction of potential defects which were the result of the sand mold system characteristics [4]. Surekha et al. (2012) used a genetic algorithm (GA) and particle swarm optimization (PSO) in the optimization of green sand mold system. After establishing a correlation between the process parameters, such as grain fineness number, percentage of clay, percentage of water and number of strokes, and responses like green compression strength, permeability, hardness and bulk density, finding an optimal solution for the green sand mold system is performed by using the above-mentioned optimization techniques. The optimization goal was to find a compromise solution that fulfills and respects all four specially considered objectives: green compression strength, permeability, hardness, and bulk density [5]. Chen et al. (2016) in their study presented a methodology that enables effective reduction in the defects produces during casting and improvement of the casting quality. Their goal was to minimize the parameters such as filling time, solidification time and oxide ratio. The optimized values were riser diameter, pouring temperatures, pouring speed, riser position and pouring diameter. They used the QPSO algorithm (improved PSO algorithm) as an optimization technique, and it enabled reducing the filling time, solidification time and oxide ratio by 68.14%, 50.56% and 20.20%, respectively [6]. Dučić et al. (2017) presented a methodology of optimization of the gating system for sand casting using the genetic algorithm (GA). The subject of optimization was the geometry of the gating system, namely the cross section of the ingate and the casting height. The main goal of optimization was to minimize the casting process time. Numerical simulation (software MAGMA5, Aachen, Germany) was used to verify the validity of the optimized geometry of the gating system [7]. Tsoukalas (2011) developed a reliable and effective methodological tool for the selection of the optimal conditions in the high-pressure die casting (HPDC) process. An adaptive neuro-fuzzy inference system (ANFIS) was applied to study the effect of die casting parameters on porosity formation in $AlSi_9Cu_3$ pressure die castings. Inputs into the developed system were: metal temperature, die temperature, piston velocity in low phase, die gate velocity and solidification pressure. After comparing the experimental results with predicted values, the author concluded that the proposed model represented an effective tool in defining optimal process conditions in pressure die-casting-associated processes, with the minimum porosity percent [8]. Zhang and Wang (2013) combined the artificial neural network and genetic algorithm (ANN/GA) in order to optimize the low-pressure die-cast (LPDC) process. The artificial neural network was used to establish a correlation between process parameters and casting part quality. The data used to build neural network models were obtained by numerical simulation of the process. The genetic algorithm was employed to optimize the process parameters with the fitness function based on the trained ANN model [9]. Zhang et al. (2012) researched the effectiveness of the genetic algorithm-based back propagation (GABP) neural network model and its application to the breakout prediction in the continuous casting process [10]. Wang et al. (2016) presented a combined approach for optimization of the cooling strategy and solidification process in continuous casting. The mentioned approach included a common application of particle swarm optimization (PSO) algorithm, mathematical heat transfer model and the experimental temperature to determine the heat transfer coefficient [11]. Sata and Ravi (2014) in their study used ANN to establish a correlation between the mechanical properties of

investment castings on one side and the process parameters and chemical composition on the other side. The data of related process parameters (wax making, shell making, dewaxing, melting etc.), the chemical composition of the alloy and the resulting mechanical properties (ultimate tensile strength, yield strength, and percentage elongation) for 800 heats were collected in an industrial investment casting foundry. Their study included the analysis of three different models of ANN (back propagation, momentum and adaptive, and Levenberg-Marquardt), and the authors concluded that ANN can successfully predict the mechanical properties [12]. Pattnaik and Kumar (2014) in their study used the genetic algorithm (GA) to improve the quality characteristic (surface finish) of the wax patterns used in the investment casting process. The optimized parameters in the surface finish function were: injection temperature, holding time and die temperature [13].

The goal of this paper is to present the application of machine learning, as a subset of artificial intelligence (AI), in the control of the white cast iron production. The control task is the control of the alloying process in order to obtain the desired chemical composition of white cast iron. The special importance of this paper is reflected in a large set of data, which was collected during three months of research in the foundry, as well as in the application of two different machine learning algorithms. In general, this study is a step towards bringing the casting process closer to the concept of the fourth industrial revolution—Industry 4.0, i.e., creating smart foundry.

## 2. Machine Learning

Machine learning (ML) is a subset of artificial intelligence (AI) which developed into a wide and diverse field of research over the past decades. The results of this are: different algorithms, theories and application areas. The most general division of machine learning algorithms is given in Figure 1.

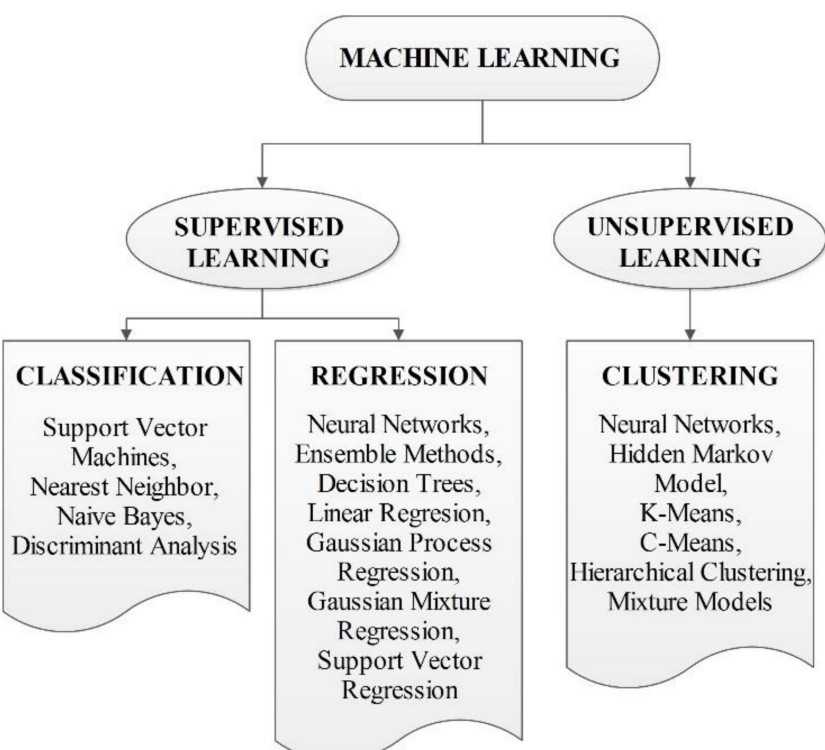

**Figure 1.** Algorithms of machine learning.

Therefore, there are many supervised and unsupervised machine learning algorithms, and each takes a different approach to learning. Choosing the right algorithm is not a simple procedure, and even experienced scientists cannot say with certainty that the algorithm will work until it is tested. The choice of the algorithm depends on the size and type of data that it is working with, as well as the result that

will be obtained from these data. The fields of application of machine learning are diverse: medical diagnosis, finance, energy production, automotive, aerospace, manufacturing, image processing and computer vision, natural language processing, etc., which clearly indicates the great potential of machine learning. Machine learning has shown a high degree of usability in solving many challenges in manufacturing systems. Machine learning algorithms are successfully used in the monitoring and optimization of production, control of manufacturing systems, as well as the predictive maintenance in different industries [14–18]. In this case of controlling the process of production of white cast iron, the most suitable for use are supervised machine learning algorithms, herein the algorithms from the category Regression. The basic principles of the functioning of the algorithms applied in this research are given below.

### 2.1. Neural Networks (NN)

Supervised neural networks are trained to precisely generate outputs from some system in response to appropriate inputs. This characteristic gives them the possibility to be applied in the modeling and control of various dynamic systems [19]. One of the most popular types of supervised neural networks are feedforward backpropagation networks. Feedforward backpropagation network can consist of a number of layers, where the first layer has a connection from the network input, and the final layer produces the network's output. The layers between the first and the last are hidden layers, and basically a feedforward network with one hidden layer and enough neurons in the hidden layers can fit any finite input-output mapping problem. For the training of a neural network, a set of data is needed that describes the process being modeled. The dataset includes network inputs $x$ and target outputs $t$. Each of the input elements, $x_1$, $x_2$, ..., $x_r$, is multiplied with the corresponding weight of the connection, $\omega_{i,1}$, $\omega_{i,2}$, ..., $\omega_{i,r}$. The neuron sums these values and adds a bias $b_i$ (lacking in some of the networks). The argument of the function (called transfer function) is stated in the following:

$$a_i = x_1\omega_{i,1} + x_2\omega_{i,2} + \ldots + x_r\omega_{i,r} + b_i \tag{1}$$

while neuron produces the output:

$$y_i = f(a_i) = f\left(\sum_{j=1}^{r} x_j\omega_{i,r} + b_i\right) \tag{2}$$

The process of training a neural network involves tuning the values of the weights and biases of the network to optimize the network performance. The principal aim is to reduce to a minimum the performance function, in this case the mean squared error (mse) function, which can be calculated as:

$$mse = \frac{1}{Q}\sum_{k=1}^{Q} e(k)^2 = \frac{1}{Q}\sum_{k=1}^{Q}(t(k) - y(k))^2 \tag{3}$$

where: $Q$—number of experiments, $e(k)$—error, $t(k)$—target values, $y(k)$—predicted values. The most commonly used training algorithm is the Levenberg–Marquardt algorithm, which ensures the fast and stable convergence. Update network weights of the Levenberg–Marquardt algorithm [20] are presented by the equation:

$$w_{k+1} = w_k - \left(J_k^T J_k + \mu I\right)^{-1} J_k e_k \tag{4}$$

and which is based on the approximation of Hessian matrix $H$.

$$H = JJ^T + \mu I \tag{5}$$

where $I$ is the identity unit matrix, $\mu$ is a learning parameter and $J$ is the Jacobian matrix (the Levenberg-Marquardt algorithm requires the computation of the Jacobian $J$ matrix at each iteration step and the inversion of $J^T J$ square matrix, the dimension of which is $NxN$).

## 2.2. Support Vector Regression (SVR)

Support vector machine (SVM) is an extremely popular machine learning technique, first introduced by Vapnik (1995) [21]. The SVM gained its popularity based on the successful application in solving problems of classification and regression. The mathematical presentation of the SVM regression algorithm, which is called the support vector regression (SVR) algorithm, is given below. The actual output of the SVR algorithm is defined by:

$$f(x) = w\varphi(x) + b \tag{6}$$

where: $\varphi(x)$—feature space, $b$—scalar, $w$—normal vector. SVR depends on a subset of the training data and it is based on minimization of the error function

$$E = \frac{1}{2}\|w\|^2 + \frac{C}{N}\sum_{i=1}^{N} L(x_i, t_i) \tag{7}$$

where: $t_i$—target value, $x_i$—input vector, $N$—data size and $\frac{C}{N}\sum_{i=1}^{N} L(x_i, t_i)$—experimental error. Minimization process is defined by the Equation (8).

$$minimize\ E(w, \xi^*) = \frac{1}{2}\|w\|^2 + \frac{C}{N}\sum_{i=1}^{N}(\xi_i, \xi_i^*)$$
$$subject\ to \begin{cases} w\varphi(x_i) + b - t_i \leq \varepsilon + \xi_i \\ t_i - w\varphi(x_i) - b \leq \varepsilon + \xi_i^* \\ \xi_i,\ \xi_i^* \geq 0,\ i = 1, 2, \ldots, N \end{cases} \tag{8}$$

where: $\frac{1}{2}\|w\|^2$—adjutstment parameters, $C$—error penalty employed to control the difference between the experimental error and adjustment parameters, $N$—number of training data sets, $\xi_i, \xi_i^*$—lower and upper allowed deviation boundaries and $\varepsilon$—loss function. The optimization problem is solved by the introduction of the Lagrangian function and Lagrange multipliers $\alpha_i, \alpha_i^*$.

$$f(x_i, \alpha_i, \alpha_i^*) = \sum_{i=1}^{N}(\alpha_i - \alpha_i^*)K(x, x_i) + b \tag{9}$$

where $K(x, x_i) = \varphi(x_i)\varphi(x_j)$ is a kernel function. The most commonly used kernel function is the radial basis function (RBF), and in addition its linear, polynomial, and sigmoid functions are also used. The RBF function is given through Equation (10).

$$K(x, x_i) = \exp\left(-\frac{\|x_i - x_j\|^2}{2\sigma^2}\right) \tag{10}$$

where: $x_i, x_j$—training and testing inputs vector and $\sigma$—Gaussian noise level.

## 3. Metal Melting: Experimental Setup and Data Collection

In the production processes in foundries, the control of the melting metals process is set as an imperative in order to obtain the desired chemical composition and temperature of the molten metal. The quantity of alloying elements and the corresponding energy sources represent the potential control objects. Metal melting (Figure 2) is a complex process because of the complex interaction between the

process parameters such as thermal loss and the dynamics of non-linear chemical reactions. In this case, the process of obtaining white cast iron was monitored and data acquisition during its monitoring was carried out. Data collection was carried out in real industrial conditions in the foundry. In the induction furnace with a capacity of eight tons, at a temperature of 1500 °C, white cast iron melts. In one shift, two batches are melted in total. The induction furnaces operate on the principle of using a strong magnetic field formed by passing the electric current through a coil, wound around the material that melts. The effect of the induced current in the metal is the heating and melting of the metal. The electromagnetic force simultaneously causes the mixing of the melted metal.

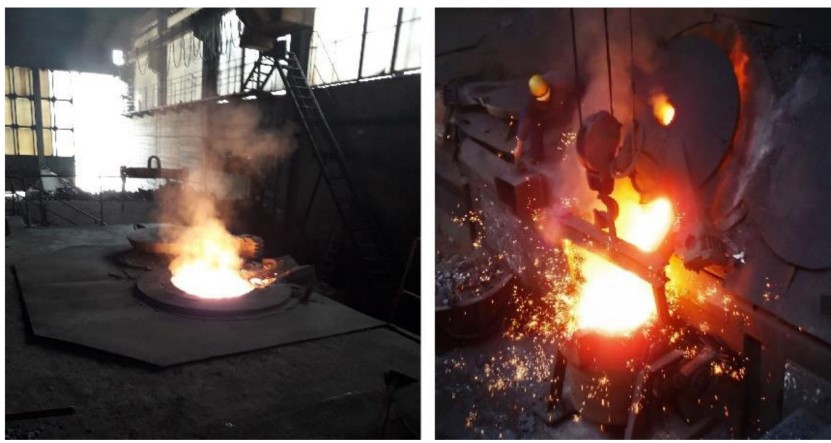

**Figure 2.** Metal melting process.

The furnace is never discharged completely during the melting process, but it remains about 30% of the total capacity, i.e., 2.5 tons of molten metal. The chemical composition of molten metal that remains in the furnace is known from the previous chemical analysis of molten metal. In the induction furnace, which has two-thirds of the empty space, it is added 5 tons of steel waste with a known chemical composition. After melting, a chemical analysis is carried out, followed by alloying if necessary, and then a new analysis and new alloying if necessary, until the desired chemical composition is achieved. Chemical analysis is carried out on a quantometer Applied Research Laboratories 2460 (Figure 3).

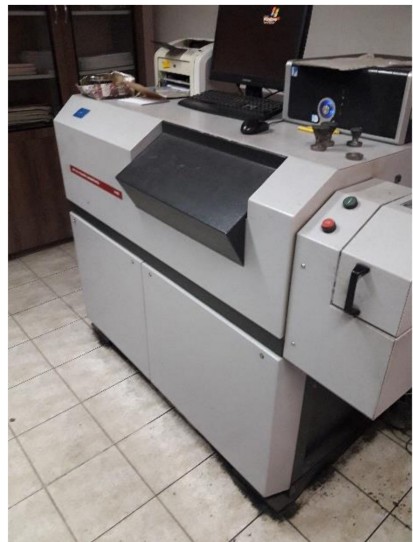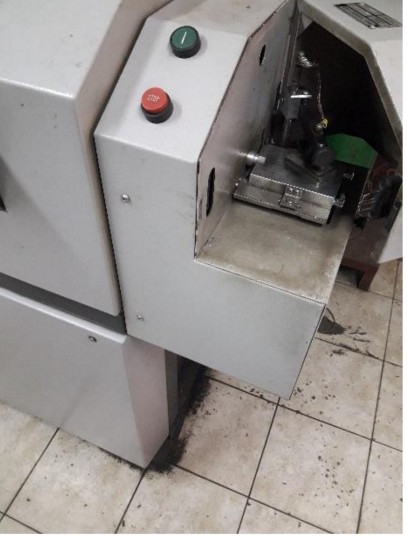

**Figure 3.** Quantometer Applied Research Laboratories 2460 for chemical analysis.

In the concrete case, four chemical elements were monitored: carbon (C), chromium (Cr), silicon (Si) and manganese (Mn), which are significant for the qualitative characteristics of the casting part for which the production of white cast iron is intended. After melting 300 batches, the same number of data sets were formed and are organized into four groups. The first group (Figure 4) consists of a chemical composition of 2.5 tons of molten metal that remains in the furnace after the melting process (C (%), Si (%), Mn (%) and Cr (%)).

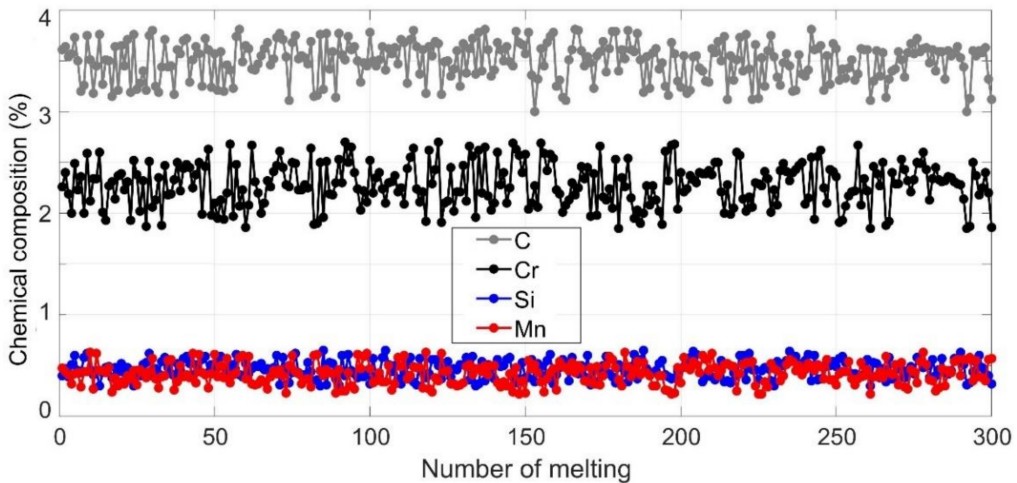

**Figure 4.** Chemical composition (C, Si, Mn i Cr) of 2.5 tons of molten metal in furnace.

The second group (Figure 5) of data consists of a chemical composition of five tons of steel waste that is added into the furnace ((C (%), Si (%), Mn (%) and Cr (%)).

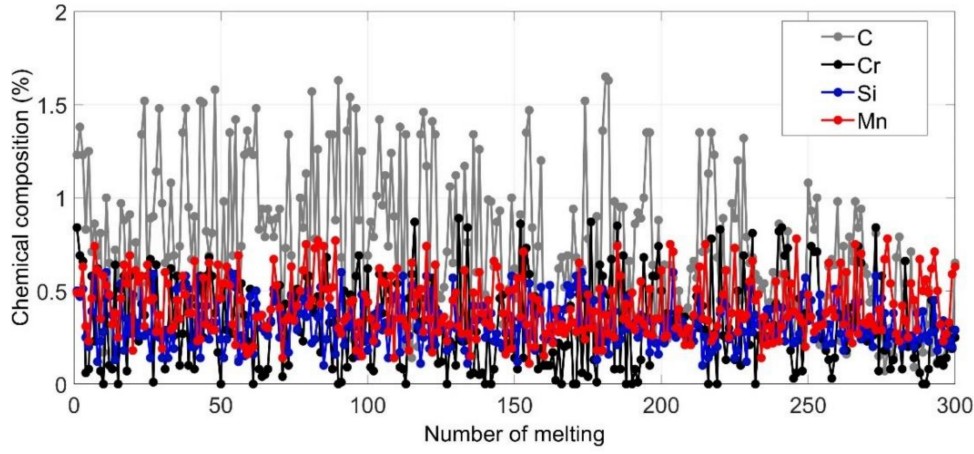

**Figure 5.** Chemical composition (C, Si, Mn and Cr) of 5 tonnes of steel waste.

The third group (Figure 6) of data represents the amount of alloying additives that is added during the alloying process (FeMn (kg), carburizing agent (kg), FeSi (kg) and FeCr (kg)).

Finally, the fourth group (Figure 7) of data represents the chemical composition of melted iron after the alloying process (C (%), Si (%), Mn (%) and Cr (%)).

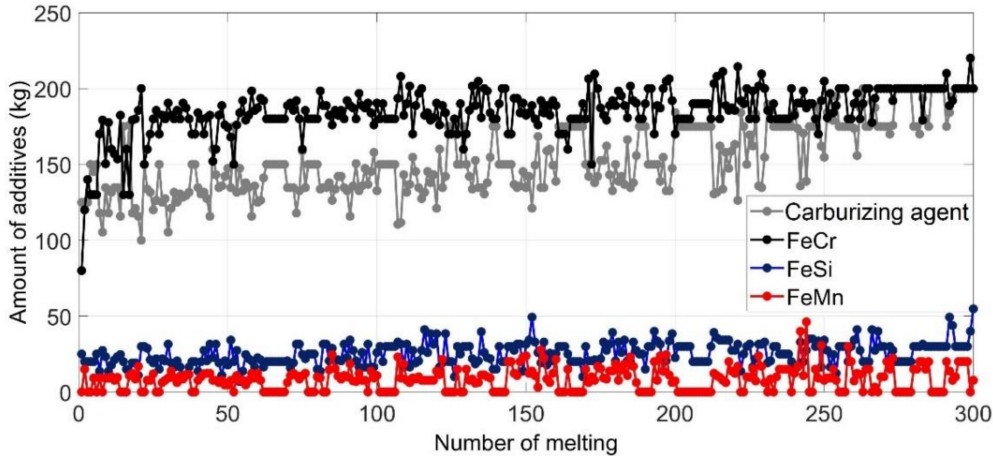

**Figure 6.** Amount (FeMn, carburizing agent, FeSi i FeCr) of alloying additives.

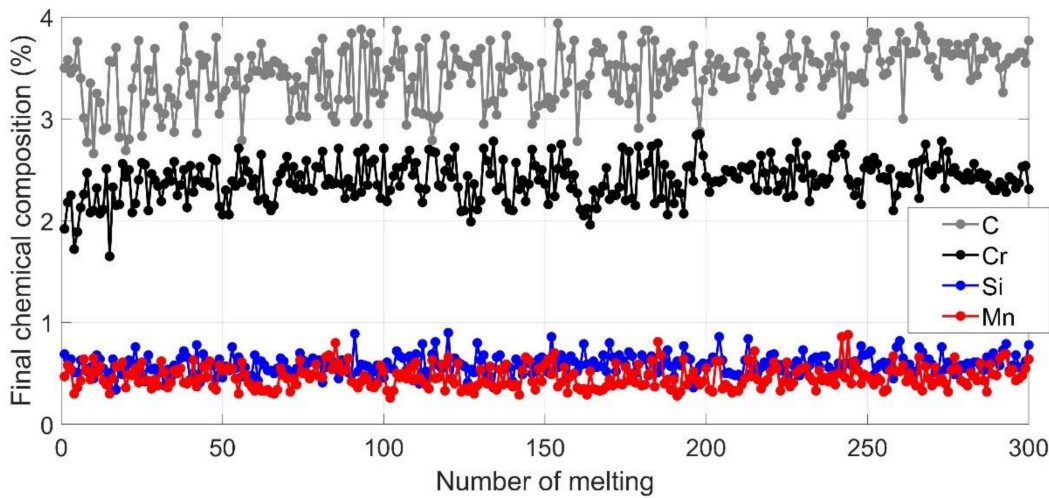

**Figure 7.** Chemical composition (C, Si, Mn and Cr) after alloying and melting.

## 4. Control of Metal Melting Process

The control of the metal melting process, with a goal to obtain the desired chemical composition of white cast iron, is based on the application of the machine learning algorithm. The machine learning algorithm carries out a prediction of the amount of alloying additives necessary to obtain the desired chemical composition of white cast iron. In accordance with the controlling idea shown in Figure 8, the data collected in the foundry are divided into the inputs and outputs for the development of the machine learning algorithm. The input dataset for the training machine learning algorithm consists of: a chemical composition of 2.5 tons of molten metal in the furnace, a chemical composition of 5 tons of steel waste and a final chemical composition of molten metal. The output dataset for the training machine learning algorithm is the amount of alloying additives that are added during the alloying process.

The application of the two described machine learning algorithms (neural networks and support vector regression) was tested. The results of the application of these two algorithms are given below.

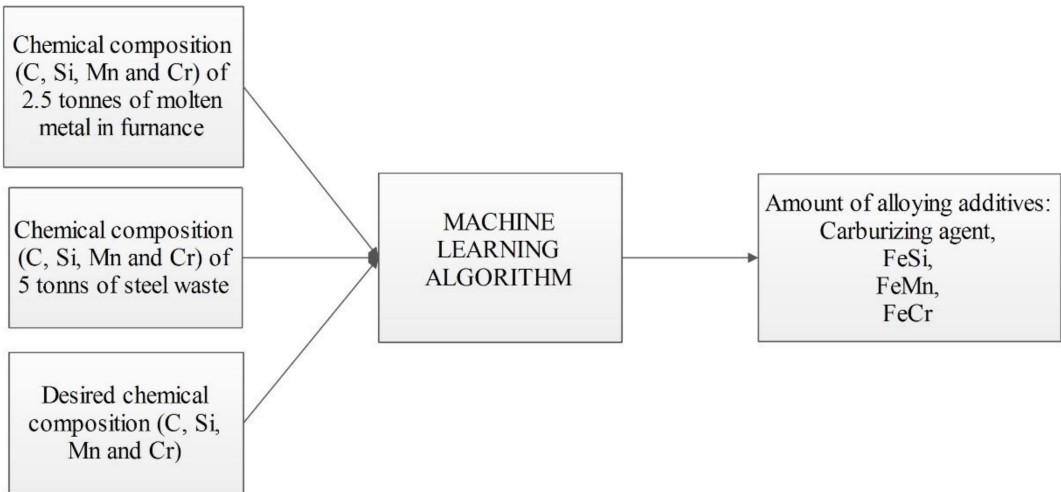

**Figure 8.** Control of white cast iron production.

### 4.1. Neural Network Control of Metal Melting Process

The above-mentioned feedforward backpropagation neural network was already created in order to control the metal melting process whose training algorithm is the Levenberg–Marquardt algorithm. The total set of collected data, that consist of 300 melting points, is divided into two groups. The first group consists of 280 data that are used for the training and development of neural network model and the second set consist of 20 data, specially selected for testing the performance of a developed neural network. The inputs into the neural network are: chemical composition of 2.5 tonnes of molten metal in the furance: 1. C (%), 2. Cr (%), 3. Mn (%), 4. Si (%), chemical composition of 5 tonnes of steel waste: 5. C (%), 6. Cr (%), 7. Mn (%), 8. Si (%) and final chemical composition of molten metal: 9. C (%), 10. Cr (%), 11. Mn (%), 12. Si (%). The output from the neural network is the amount of alloying additives: carburizing agent (kg), FeCr (kg), FeMn (kg) and FeSi (kg). In the training phase, a set of 280 data was divided into the training dataset, the validation dataset and the test dataset in this percentage relation—80%:10%:10%. Several different neural network architectures were designed with a variety of hidden layers and neurons in them. Table 1 provides different neural network architectures and corresponding mean square errors in the training phase

**Table 1.** Mean squared errors in the training phase— neural network (NN) models.

| NN Model (Input-Hidden-Output) | Mean Squared Error |
|:---:|:---:|
| 12-14-4 | 0.23 |
| 12-15-4 | 0.31 |
| 12-18-4 | 0.28 |
| 12-16-4 | 0.16 |
| 12-12-4 | 0.25 |

In Figure 9 is shown a neural network architecture, which showed the best performances.

This neural network (NN12-16-4) has one hidden layer with 16 neurons in it, the input layer is defined by the number of inputs (12), and output layer is defined by the number of outputs (4). The neurons in input and hidden layers of neural networks had the sigmoid transfer function, while the neurons of the output layer have the linear transfer function. The mean squared error in the training phase, as a very significant performance of neural network (NN12-16-4), is 0.16. As a validity measure of a neural network, maximum and the mean errors in the test phase are used. The testing has been performed with a set of 20 data that did not participate in the development of the neural network model. The maximum and mean errors for the prediction of four alloying additives are given in Table 2.

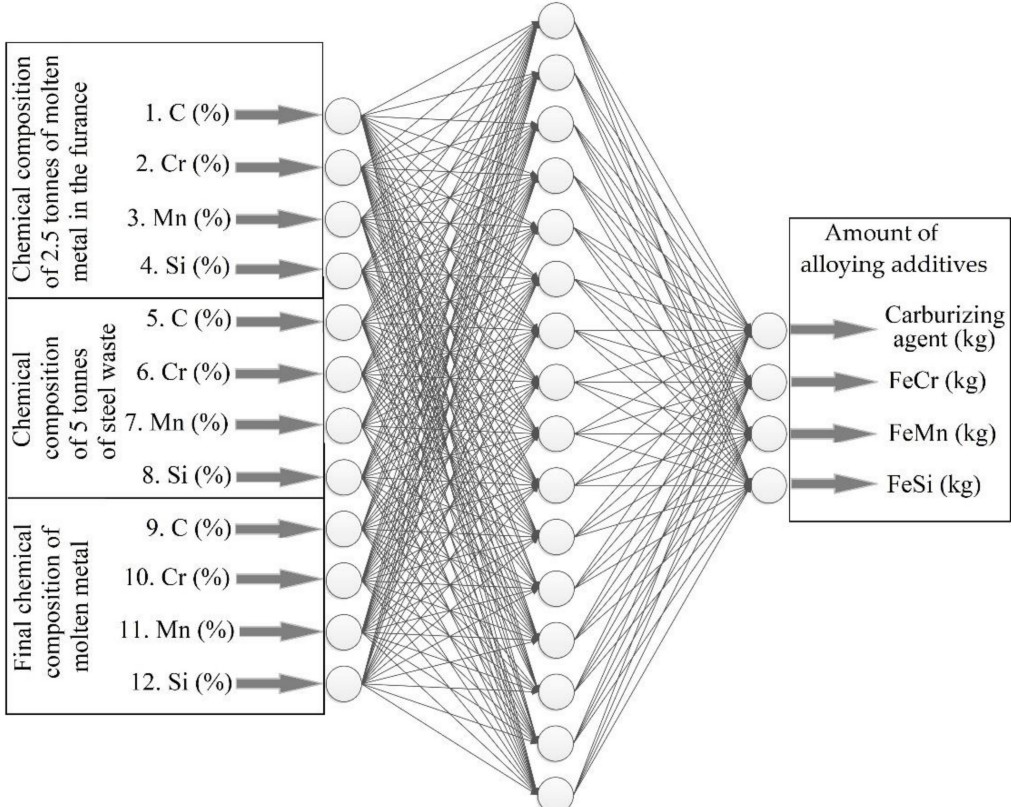

**Figure 9.** Neural network architecture for prediction of the amount of alloying additives.

**Table 2.** Maximum and mean error in the testing phase of the neural network.

|  | Carburizing Agent (kg) | FeCr (kg) | FeMn (kg) | FeSi (kg) |
|---|---|---|---|---|
| Mean error (%) | 0.47 | 0.51 | 3.31 | 1.88 |
| Max. error (%) | 1.97 | 1.92 | 9.42 | 8.18 |

*4.2. Support Vector Regression Control of the Metal Melting Process*

Support vector regression (SVR) control of the metal melting is based on the development of four SVR models, which simultaneously perform the prediction of the alloying additives amount (Figure 10). Each of the developed models performs the prediction of the amount of one alloying additive. As well as in the case of the neural network development, a set of 280 data is used for the development of SVR algorithms, and a set of 20 data, which does not participate in development of algorithms, is used for testing of the developed SVR models. The inputs into SVR algorithms are: chemical composition of 2.5 tons of molten metal in the furnace: 1. C (%), 2. Cr (%), 3. Mn (%), 4. Si (%), chemical composition of 5 tons of steel waste: 5. C(%), 6. Cr(%), 7. Mn(%), 8. Si(%) and final chemical composition of molten metal: 9. C (%), 10. Cr (%), 11. Mn (%), 12. Si (%). The output from each algorithm is different and represents one of the alloying additives: carburizing agent (kg), FeCr (kg), FeMn (kg) and FeSi (kg).

The developed SVR models are based on the use of the RBF kernel function. RBF, as the kernel function for SVR, has become the choice of researchers because of its accuracy and reliable performance. The parameters C from Equation (8) and σ from Equation (10) significantly affect the prediction potential and need to be carefully determined. The optimum values of *C* and *σ* parameters were determined by using the grid-based search technique proposed by Hsu et al. (2003) [22]. Table 3 provides the parameter values and the corresponding mean square error for each model in three cases.

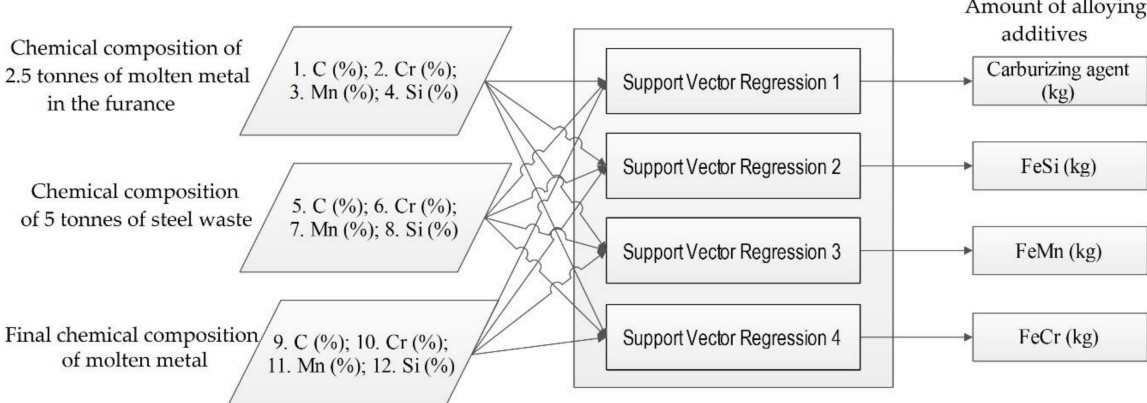

**Figure 10.** Support vector regression (SVR) control strategy for the prediction of the amount of alloying elements.

**Table 3.** Mean squared errors in the training phase—SVR models.

|  | Models | $C$ | $\sigma$ | Mean Squared Error (mse) |
|---|---|---|---|---|
| Case 1 | Support Vector Regression 1 | 6 | 0.02 | 0.88 |
|  | Support Vector Regression 2 | 6 | 0.02 | 0.98 |
|  | Support Vector Regression 3 | 6 | 0.02 | 0.75 |
|  | Support Vector Regression 4 | 6 | 0.02 | 0.65 |
| Case 2 | Support Vector Regression 1 | 10 | 0.005 | 0.45 |
|  | Support Vector Regression 2 | 10 | 0.005 | 0.47 |
|  | Support Vector Regression 3 | 10 | 0.005 | 0.42 |
|  | Support Vector Regression 4 | 10 | 0.005 | 0.38 |
| Case 3 | Support Vector Regression 1 | 32 | 0.12 | 0.84 |
|  | Support Vector Regression 2 | 32 | 0.12 | 1.13 |
|  | Support Vector Regression 3 | 32 | 0.12 | 0.95 |
|  | Support Vector Regression 4 | 32 | 0.12 | 0.48 |

Based on the training phase, the greatest prediction potential was shown by case 2 when the parameters $C$ and $\sigma$ had values of 10 and 0.005, respectively. In this case, the mean squared errors in the training phase for four SVR models are in range 0.38–0.47. As a measure of validity of the SVR models, the maximum and mean error in the testing phase was used. The testing is performed with a set of 20 data, and the maximum and mean error for prediction four alloying additives are given in Table 4.

**Table 4.** Maximum and mean error in the testing phase of SVR.

|  | Carburizing Agent (kg) | FeCr (kg) | FeMn (kg) | FeSi (kg) |
|---|---|---|---|---|
| Mean error (%) | 3.39 | 2.05 | 4.81 | 6.65 |
| Max. error (%) | 9.29 | 6.77 | 10.57 | 14.56 |

### 4.3. Discussion of Results

Comparing the results obtained by the two types of supervised machine learning algorithms, it can be concluded that the neural network model has given a better response and an excellent result in the testing phase (Figures 11–14). This is exactly what qualifies the NN model to be implemented into the control structure of the white cast iron production. However, it should also be noted that SVR models demonstrate a high level of success in the prediction of alloying additives.

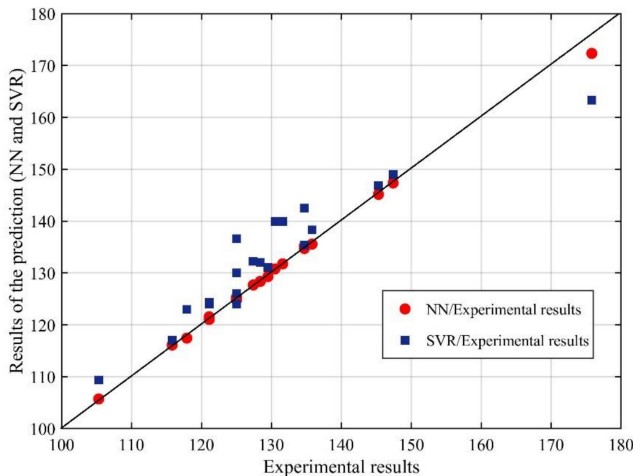

**Figure 11.** Results of the prediction for alloying additive carburizing agent.

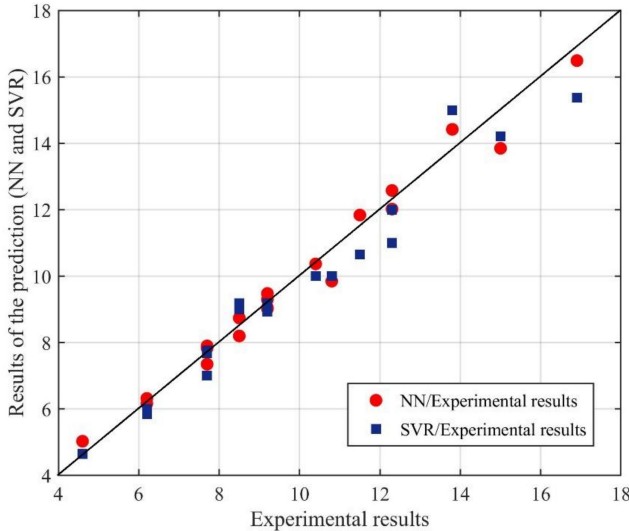

**Figure 12.** Results of the prediction for alloying additive FeMn.

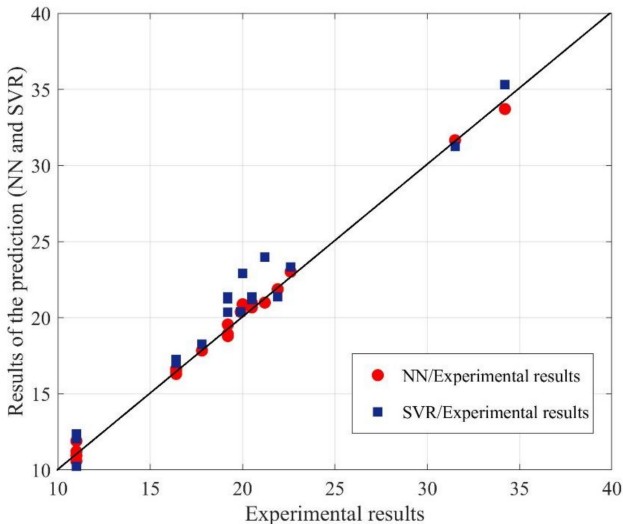

**Figure 13.** Results of the prediction for alloying additive FeSi.

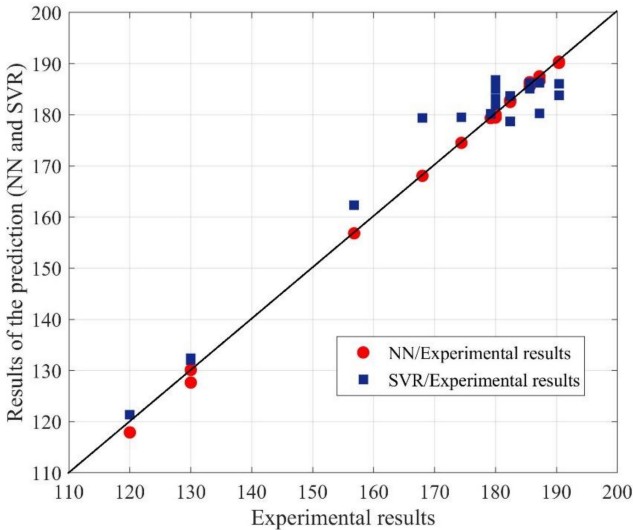

**Figure 14.** Results of the prediction for alloying additive FeCr.

Carburizing agent: In the testing phase, the NN model provides a smaller mean and maximum errors compared to the SVR model. The mean and maximum errors of the NN model is 0.47 and 1.97%, respectively. The mean and maximum errors of the SVR model is 3.39 and 9.29%, respectively.

FeMn: In the testing phase, the NN model provides a smaller mean and maximum error compared to the SVR model. The mean and maximum errors of the NN model are 3.31% and 9.42%, respectively. The mean and maximum errors of the SVR model are 4.81% and 10.57%, respectively.

FeSi: In the testing phase, the NN model provides smaller mean and maximum errors compared to the SVR model. The mean and maximum errors of the NN model are 1.88% and 8.18%, respectively. The mean and maximum errors of the SVR model are 6.65% and 14.56%, respectively.

FeCr: In the testing phase, the NN model provides smaller mean and maximum errors compared to the SVR model. The mean and maximum errors of the NN model are 0.51% and 1.92%, respectively. The mean and maximum errors of the SVR model are 2.05% and 6.77%, respectively.

If the maximum error of the SVR model is observed, which is the highest in the case of the FeSi additive (14.56%), the difference between the experimental value and the value obtained by the SVR algorithm is 2.91 kg. The maximum error of the NN model is in the case of the prediction of the FeMn additive (9.42%), and the difference between the experimental value and the value given by the NN model is only 400 g. Thus, both models provide small deviations from experimental value, which do not pose a threat for the final chemical composition of white cast iron.

## 5. Conclusions

The subject of this study was to control the metal melting process. Metal melting is a very complex process because of the complex interaction between process parameters such as thermal loss and the dynamics of non-linear chemical reactions. The complexity and nonlinearity of this production process were a research challenge for the application of machine learning algorithms in its improvement.

The paper presents the application of two supervised machine learning algorithms in the control of the alloying process during the white cast iron production. The models of the neural network (NN) and the support vector regression (SVR) algorithms have been developed to perform the prediction of the amount of alloying additives in order to obtain a desired chemical composition of white cast iron. For the development of models and their testing, a set of 300 data was utilized, which were collected in the foundry during the three-month-long monitoring of the metal melting process. Several different NN and SVR models were developed, and the models with the best performance (mean squared error) were selected. Selected models were tested on an unknown data set. The testing results show that both the NN and SVR models were suitable and reliable for the control of the alloying process during the

white cast iron production. The NN model showed greater accuracy in the prediction of all alloying additives (FeMn, carburizing agent, FeSi i FeCr). We achieved the worst result in the prediction of the alloying additive FeMn, when the mean error was 3.31% and the maximum error was 9.42%. The SVR model achieved the worst result in the prediction of the FeSi alloying additive, when the mean error was 6.65% and the maximum error was 14.56%. The superior accuracy of the NN model suggests that it could be a very powerful tool for the control of activities at the foundry regarding the metal melting process.

The presented control systems based on the two supervised machine learning algorithms are methodologically applicable to a wide range of alloys that are obtained in the melting process. Their application positively influences the precision and efficiency of the production, and this is in the accordance with the idea of designing smart foundries as a segment of the concept of Industry 4.0.

**Author Contributions:** Conceptualization: N.D., S.M. and Ž.Ć. Methodology: N.D., Ž.Ć. and B.S. Resources: R.R. and A.J. Data curation: R.R., A.J. and S.M. Software: N.D. and B.S. Writing—original draft preparation: N.D. All authors have read and agreed to the published version of the manuscript.

**Funding:** This research received no external funding.

**Acknowledgments:** This study was supported by the Ministry of Education, Science and Technological Development of the Republic of Serbia.

**Conflicts of Interest:** The authors declare no conflict of interest.

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
