# Peer review of "Application of Machine Learning in the Control of Metal Melting Production Process"

_applsci, doi:10.3390/app10176048_

Round 1

Reviewer 1 Report

Review of applsci-893075-peer-review-v1: “Application of machine learning in the control of metal melting production process”

The subject of the paper is relevant with the topics of the journal.

The number and the quality of the references selected are adequate while their use is correct.

Its significance with respect to their industrial value is very good and its basis on a great number of experimental works proves to be solid.

I would suggest the authors to incorporate the following in order to improve the quality of the paper:

  • Although figures 4, 5, 6 and 7 are well presented, a detailed table with all the numbers used, would offer a great deal of help for future researchers and increased citation activities of the article.
  • In figure 9, separation of the input data based on the set of input used will improve the graph.
  • In lines 265-266: “Mean squared error in the training phase, as a very significant performance of neural network (NN12-16-4) is 0.16”. The authors should state in a table, what are the results in other configurations tried i.e. NN12-15-4.
  • In lines 284-285, similar comments of the work performed should be included for SVR as well.

My proposal to the editor is to accept the paper after minor revisions.

Author Response

Dear reviewer,

Thanks for the suggestions, your advice is of great importance for improving the quality of this paper.

1. We agree that the presentation of numerical data would be of great importance, but a table with all sets of experimental data would occupy 8 pages of our paper. We think that would be a great burden for the reader. Of course, we can send you a spreadsheet, if you wish.

2. We have accepted the advice and Figure 9 has been corrected (line in text 265 – red text), we have also applied your advice in Figure 10 (line in text 288 – red text).

3. Our main goal was to compare the success of SVR and NN in the metal melting process control. Therefore, the best models of the two machine learning techniques were singled out. We accept your suggestion and in Table 1 (line in text 262– red text) and Table 3 (line in text 297– red text) we have singled out some more models of two machine learning techniques.

Best Regards,

Authors

Reviewer 2 Report

It seems unfeasible to train and test a neural network against the same data set. It is suggested that the NN training should be performed against available online datasets and then tested against the foundry datasets.

Author Response

Dear reviewer,

First of all, thank you for reviewing our paper.
We made many corrections based on the received suggestions.

Our experimental data set consists of 300 data, each of which represents one melting of metal in a foundry. Process monitoring and data collection took about three months.

Data sets (300) are divided into two groups. The first group consists of 280 data sets and the second of 20 data sets. The first data group (280) is used to design neural networks. Lines 252-254 in the text of the paper.

And another set of data (20) is used to test the neural network. In our case it is the NN12-16-4 network. This data set did not participate in the development of the neural network. The data used in the testing phase are absolutely unknown to the neural network. Line 272-274 in the text of the paper.

The same principle was applied to the SVR model, 280 data sets to create the model, and 20 data sets to test its performance.

Figures 11-14 present a comparison of the results of the NN and SVR models in the testing phase (20 data).

If you need another explanation, we will be happy to answer you.

Best regards,
Authors

Reviewer 3 Report

  • The manuscript is in good form. However, minor modifications required.
  • Please double check the English writing part. There are some spelling mistakes.
  •  Conclusions need to be strengthen with more detailed ellaboration. 

Author Response

Dear reviewer,

First of all, thank you for reviewing our paper.

1. We performed the checks you suggested.

2. The conclusion of the paper is expanded (line in text 342-346; 352-363 – green text). We highlighted the results and promoted the Neural network model as a software tool ready for industrial use (line in text 352-363– green text). Details are also given that clearly show the idea of the paper and its correlation with Industry 4.0.

Thanks again and best regards,
Authors

Reviewer 4 Report

The authors applied two machine learning algorithms to optimise the alloying during the melting process of cast iron. The research is of interest for the journal and the casting industry in general. On the positive side: The training of the SVN and NN is performed on a data set which is clearly seperated from testing and validation data sets. However, I would like to see more on the developement of the NN. How sensitive are the results to changes in the architecture of the NN?   More importantly: The paper merely describes the results of the training. A discussion of the results is completely missing. Furthermore a conclusion should not only summarise the results of the paper, but also discuss them and their significance in a more abstract context.   Minor Issue: There are some spelling mistakes in the manuscript, which should be improved.

Author Response

Dear reviewer,

First of all, thank you for reviewing our work, and thank you for your useful suggestions.

1. We have accepted your suggestion and presented the influence of different NN architectures on the mean square error. Table 1 (text line 261-262 – red text).
By the same principle for the SVR algorithm, we presented the influence of the values of important parameters of the algorithm on the mean square error. Table 3 (text line 292-298 – red text).

2. The fourth chapter of the paper is expanded with subchapter 4.3. Discussion of results (text line 305-340 – blue text). In that part of the paper, for each alloying additive, in addition to the graphical representation, a comment is given on the results of the ML algorithms used in the testing phase. 

3. The conclusion of the paper is expanded (line in text 342-346; 352-363 – green text). We highlighted the results and promoted the Neural network model as a software tool ready for industrial use. Details are also given that clearly show the idea of the paper and its correlation with Industry 4.0.

4. We checked the text of the paper.

Thanks again and best regards,
Authors

Round 2

Reviewer 2 Report

Some grammatical and textual issues remain, such as

Line 26: "The casting process is an old manufacturing technology whose fundamental principles have not been changed up to this day." should be

"The casting process is an old manufacturing technology of which the fundamental principles have not changed to this day."

Line 57: "Surekha et al. (2012) used A genetic algorithm (GA) and particle swarm optimization (PSO)" should be

"Surekha et al. (2012) used a genetic algorithm (GA) and particle swarm optimization (PSO)"

Everywhere: "green sand mold system" should be

"green sand mould system"

Author Response

Dear reviewer,   1. We made the correction - line 26 (blue text). 2. We made the correction - line 57 (blue text). 3. We made the correction - “green sand mould system” in lines 56,58 and 61 (blue text).   Thank you for reviewing our paper.
Best regards,
Authors

Reviewer 4 Report

The authors have adressed our issues.

Author Response

Dear reviewer,    thank you for reviewing our paper.   Best regards,
Authors